# A novel approach to explore common prime divisor graphs and their degree based topological descriptor

**Ali N. A. Koam**[ID]**[1], Azeem Haider**[ID]**[1]\*, Ali Ahmad[2], Moin Akhtar Ansari**[ID]**[1]\***

**1** Department of Mathematics, College of Science, Jazan University, Jazan, Kingdom of Saudi Arabia,
**2** Department of Computer Science, College of Engineering and Computer Science, Jazan University, Jazan, Kingdom of Saudi Arabia

\* aahaider@jazanu.edu.sa (AH); maansari@jazanu.edu.sa (MAA)

**Data availability statement:** All relevant data are within the paper.

## Abstract

For the construction of a common prime divisor graph, we consider an integer $\zeta = \prod_{i=1}^{k} p_i^{\gamma_i} \geq 2$ with its prime factorization, where $p_i's$ are distinct primes and $\gamma_i's$ are fixed positive integers. Every divisor of the integer $\zeta$ has the form $x = \prod_{i=1}^{k} p_i^{x_i}$, with $0 \leq x_i \leq \gamma_i$. There are $\prod_{i=1}^{k} (\gamma_i + 1)$ distinct divisors of integer $\zeta$. Let $D(\zeta)$ be the collection of all positive divisors of $\zeta$ other than integer 1. Then we can define a simple graph on the set of divisors $D(\zeta)$ of $\zeta$, called a *common prime divisor graph* $\mathfrak{Z}(\zeta)$ with $D(\zeta)$ as the vertex set, and we insert an edge between two distinct divisors $x$ and $y$ of $\zeta$ if the $\gcd(x, y) = p_i$. In this article, we will introduce and discuss some basic properties of common prime divisor graphs and we will compute some indices of symmetries associated with a class of such graphs. This study will open a new domain of graphs to investigate their invariant and to explore such indices on the different classes of common prime divisor graphs.

## Introduction

Topological indices represent an integral aspect of computational chemistry and provide quantitative descriptors for molecular structures that facilitate the analysis of complex chemical relationships. These indices, rooted in graph theory, offer insights into molecular connectivity without considering spatial arrangement. This summary will delve into the foundational contributions, diverse applications, and recent advancements in topological indices, referencing the key literature [1–6]. Some geometrical properties of annihilator intersection graph of commutative rings are explored in [7] and the embedding of the extended zero-divisor graph of commutative rings are explored in [8].

A pioneering topological index, the Wiener index, was initiated by Harry Wiener in 1947. In fact, this index is the sum of the pairwise shortest distances of paths between atoms in a given molecular graph, offering a calculation of molecular size with its complexity. Wiener, in his seminal paper, "Structural Determination of Paraffin Boiling Points," laid the groundwork for subsequent developments in topological indices by demonstrating the significance of graph-based descriptors for understanding molecular properties [9].

**Funding:** The authors gratefully acknowledge the funding of the Deanship of Graduate Studies and Scientific Research, Jazan University, Saudi Arabia, through project number: RG24-L04. The funders had no role in study design, data collection and analysis, decision to publish, or preparation of the manuscript.

**Competing interests:** The authors have declared that no competing interests exist.

As the field progressed, various topological indices emerged, to capture different aspects of the molecular structure. In 1975, the Randic index, introduced by Milan Randic, concentrated on the balance between high- and low-degree nodes in a molecule, providing a different perspective on its structural features [10]. Additionally, indices such as Zagreb indices [11,12], eccentricity based indices [13,14], distance based indices [15], Sombor indices [16,17], M-polynomial [18] and resistance-based indices [19] contribute to a comprehensive characterization of the molecular structure [20–23].

Topological indices have various application across scientific disciplines. For example, in medicinal chemistry, these indices are used for a fundamental index, that is, "Quantitative Structure-Activity Relationship", also called "QSAR". "Molecular Descriptors for Chemoinformatics," introduced by Todeschini-Consonni, that are used to explore the applications of topological indices modeling of QSAR, has a key role for predicting the toxicity or bioactivity of chemical compounds [24]. Using these structures, researchers have managed to unravel the relationships amongst the molecular structure and its biological activity and, provide the fundamental design of drugs [25,26].

In particular, degree based topological indices have significant applications including drug design, modeling of quantitative structure-activity relationships (QSARs), and molecular similarity analysis. A rapid and effective technique was developed to compare the connectivity of atoms in various compounds. Moreover, these indices are comparatively easy to compute and compare [27,28]. Chemical species and chemical processes can be represented as nodes and edges in a graph, which can then be used to depict a chemical network. Chemical reactions link the reactants and products of the reaction at their respective nodes. This enables the chemical network to be described as a directed graph, where each node represents a particular chemical species and each edge for a particular chemical process [29–31].

Owing to the extensive interest in topological concepts related to graphs for ring structures, many researchers have shown their interest in indices studied on zero-divisor graphs defined over rings [15,32–34].

Topological indices are useful in chemoinformatics for molecular similarity analysis, clustering, and virtual screening. The work of Dehmer and Varmuza in "Advances in Quantitative Structure-Property Relationships" comprehensively discusses the role of topological indices in chemoinformatics, showcasing their versatility in handling large chemical datasets [35].

## 1 Preliminaries

Consider a connected simple graph $\mathfrak{S}$ with $\mathcal{V}(\mathfrak{S})$ and $\mathcal{E}(\mathfrak{S})$ representing the set of vertices and edges of $\mathfrak{S}$ respectively. Todesehini *et al.* [36] defined ${}^t\phi_{\mathfrak{S}}(\alpha) = |{}^tN_{\mathfrak{S}}(\alpha)|$ for ${}^tN_{\mathfrak{S}}(\alpha) = \{\beta \in \mathcal{V}(\mathfrak{S}) : d(\alpha,\beta) = t\}$ such that ${}^1N_{\mathfrak{S}}(\alpha) = d_{\mathfrak{S}}(\alpha)$ is named the degree of a vertex $\alpha \in \mathcal{V}(\mathfrak{S})$ and ${}^2N_{\mathfrak{S}}(\alpha) = \aleph_{\mathfrak{S}}(\alpha)$ are connection number of the vertex $\alpha \in \mathfrak{B}(\mathfrak{S})$.

We define the $\mathfrak{R}_\alpha$ index or the general Randić index:

$$\mathfrak{R}_\alpha(\mathfrak{S}) = \sum_{\gamma\beta \in \mathfrak{E}(\mathfrak{S})} (d_{\mathfrak{S}}(\gamma) \times d_{\mathfrak{S}}(\beta))^\alpha \tag{1}$$

where $\alpha$ is a real number and for $\alpha = -\frac{1}{2}, -1, 1$, we first obtain the Randić index, then the second modified Zagreb index, and finally second Zagreb index. Here we define $\chi_\alpha$ index or general sum-connectivity index;

$$\chi_\alpha(\mathfrak{S}) = \sum_{\gamma\beta \in \mathfrak{E}(\mathfrak{S})} (d_{\mathfrak{S}}(\gamma) + d_{\mathfrak{S}}(\beta))^\alpha \tag{2}$$

where $\alpha$ is a real number and for $\alpha = -\frac{1}{2}, 1, 2$, we get firstly the sum-connectivity, then the first Zagreb index and lastly the hyper-Zagreb index.

We define the *GA* (geometric-arithmetic) index:

$$GA(\mathfrak{S}) = \sum_{\gamma\beta\in\mathfrak{E}(\mathfrak{S})} \frac{2\sqrt{d_{\mathfrak{S}}(\gamma) \times d_{\mathfrak{S}}(\beta)}}{d_{\mathfrak{S}}(\gamma) + d_{\mathfrak{S}}(\beta)} \tag{3}$$

We define the *ABC* (atom bond connectivity) index as:

$$ABC(\mathfrak{S}) = \sum_{\gamma\beta\in\mathfrak{E}(\mathfrak{S})} \sqrt{\frac{d_{\mathfrak{S}}(\gamma)+d_{\mathfrak{S}}(\beta)-2}{d_{\mathfrak{S}}(\gamma)\times d_{\mathfrak{S}}(\beta)}}. \tag{4}$$

We define the *AZI* (augmented Zagreb) index as:

$$AZI(\mathfrak{S}) = \sum_{\gamma\beta\in\mathfrak{E}(\mathfrak{S})} \left(\frac{d_{\mathfrak{S}}(\gamma) \times d_{\mathfrak{S}}(\beta)}{d_{\mathfrak{S}}(\gamma) + d_{\mathfrak{S}}(\beta) - 2}\right)^3. \tag{5}$$

We define the *H* (harmonic) index as:

$$H(\mathfrak{S}) = \sum_{\gamma\beta\in\mathfrak{E}(\mathfrak{S})} \frac{2}{d_{\mathfrak{S}}(\gamma) + d_{\mathfrak{S}}(\beta)}. \tag{6}$$

We define the *SDD* (symmetric division degree) index as:

$$SDD(\mathfrak{S}) = \sum_{\gamma\beta\in\mathfrak{E}(\mathfrak{S})} \frac{(d_{\mathfrak{S}}(\gamma))^2 + (d_{\mathfrak{S}}(\beta))^2}{d_{\mathfrak{S}}(\gamma) \times d_{\mathfrak{S}}(\beta)}. \tag{7}$$

We define the $ReZG_1$ (first redefined Zagreb) index as:

$$ReZG_1(\mathfrak{S}) = \sum_{\gamma\beta\in\mathfrak{E}(\mathfrak{S})} \frac{d_{\mathfrak{S}}(\gamma) + d_{\mathfrak{S}}(\beta)}{d_{\mathfrak{S}}(\gamma) \times d_{\mathfrak{S}}(\beta)}. \tag{8}$$

We define the $ReZG_2$ (second redefined Zagreb) index as:

$$ReZG_2(\mathfrak{S}) = \sum_{\gamma\beta\in\mathfrak{E}(\mathfrak{S})} \frac{d_{\mathfrak{S}}(\gamma) \times d_{\mathfrak{S}}(\beta)}{d_{\mathfrak{S}}(\gamma) + d_{\mathfrak{S}}(\beta)}. \tag{9}$$

We define the $ReZG_3$ (third redefined Zagreb) index as:

$$ReZG_3(\mathfrak{S}) = \sum_{\gamma\beta\in\mathfrak{E}(\mathfrak{S})} (d_{\mathfrak{S}}(\gamma) \times d_{\mathfrak{S}}(\beta)) (d_{\mathfrak{S}}(\gamma) + d_{\mathfrak{S}}(\beta)). \tag{10}$$

## 2 Main results

Consider a positive integer $\zeta > 1$ with the prime factorization $\zeta = \prod_{i=1}^{k} p_i^{\gamma_i}$, where $p_i's$ are distinct prime numbers and each $\gamma_i$ is a fixed positive integer. Any divisor $x$ of $\zeta$ has the form

$x = \prod_{i=1}^{k} p_i^{x_i}$, where $0 \le x_i \le \gamma_i$. Note that, the integer $\zeta$ has $\prod_{i=1}^{k}(\gamma_i + 1)$ number of distinct divisors.

Let $D(\zeta) = \{x = \prod_{i=1}^{k} p_i^{x_i} \mid x_i \le \gamma_i, \; x > 1\}$ be the set of all positive divisors of $\zeta$. Then we define a graph on the set of divisors of $\zeta$, called a *common prime divisor graph* $\Im(\zeta)$ with the vertex set $D(\zeta)$ with an edge between any two divisor $x = \prod_{i=1}^{k} p_i^{x_i}$ and divisor $y = \prod_{i=1}^{k} p_i^{y_i}$ if $\gcd(x, y) = p_i$ for any $i \in \{1, 2, 3, \dots, k\}$.

One can easily see that the order of the graph $\Im(\zeta)$ is $|D(\zeta)| = \prod_{i=1}^{k}(\gamma_i + 1) - 1$.

**Lemma 2.1.** *Let $\zeta = \prod_{i=1}^{k} p_i^{\gamma_i}$ be a positive integer. Then for any two vertices $x = \prod_{i=1}^{k} p_i^{x_i}$ and $y = \prod_{i=1}^{k} p_i^{y_i}$ of the common prime divisor graph $\Im(\zeta)$, the distance $d(x, y) = 1$ or the distance $d(x, y) = 2$. Moreover, if $|D(\zeta)| \ge 3$, then the diameter, $\mathrm{diam}(\Im(\zeta)) = 2$.*

*Proof*: If $(x, y) = p$ for any prime number $p$, then $d(x, y) = 1$. If $(x, y) = \prod_{i=1}^{k} p_i^{z_i}$ such that $\sum_{i=1}^{k} z_i \ge 2$, then for any prime $p \mid (x, y)$ we obtained a path $x$–$p$–$y$ implies that the distance $d(x.y) = 2$. If $(x, y) = 1$, then for a prime $p$ and a prime $q$, we see that if $p$ divides $x$ and $q$ divides $y$, then we have a path $x$–$pq$–$y$ and hence $d(x, y) = 2$.

Moreover, if $k = 1$ and $|D(\zeta)| \ge 3$, then we have, $\zeta = p^\alpha$, $\alpha \ge 3$. Clearly in this case, $d(p^2, p^3) = 2$ and if $k \ge 2$, then $d(p_1, p_2) = 2$. □

For any divisor $x = \prod_{i=1}^{k} p_i^{x_i}$ of $\zeta = \prod_{i=1}^{k} p_i^{\gamma_i}$, $\gamma_i > 0$, we partition the set of indices $I = \{1, 2, 3, \dots, k\}$ for $x$ into three sets $A_x = \{j \mid x_j = 0\}$, $B_x = \{j \mid x_j = 1\}$ and $C_x = \{j \mid x_j \ge 2\}$. This partition will help us to construct the following results on the degrees of the common prime divisor graph $\Im(\zeta)$.

**Theorem 2.2.** *Let $\zeta = \prod_{i=1}^{k} p_i^{\gamma_i}, \gamma_i > 0$ be a positive integer. Then in the graph $\Im(\zeta)$.*

(a) $\deg(p_j) = \left( \gamma_j . \prod_{i \in I \setminus \{j\}} (\gamma_i + 1) \right) - 1.$

(b) $1 < t \le \gamma_j, \deg(p_j^t) = \prod_{i \in I \setminus \{j\}} (\gamma_i + 1).$

(c) *For any $x = \prod_{i=1}^{k} p_i^{x_i}$ with $|A_x| \le k - 2$, $\deg(x) = \left( \sum_{j \in B_x} \gamma_j + |C_x| \right) . \prod_{i \in A_x} (\gamma_i + 1).$*

*Proof*:  (a) The vertex $p_j$ is connected to $y \ne p_j$ if and only if $y = \prod_{i=1}^{k} p_i^{y_i}$ for each $y_j \in \{1, 2, \dots, \gamma_j\}$ and if $i \in I \setminus \{j\}$, $y_i \in \{0, 1, 2, \dots, \gamma_i\}$. Since $y \ne p_j$, therefore we exclude the case if $y_i = 0$ for all $i \in I \setminus \{j\}$. Clearly, $\deg(p_j) = \left( \gamma_j . \prod_{i \in I \setminus \{j\}} (\gamma_i + 1) \right) - 1.$

(b) The vertex $p_j^t$ for $t \in \{2, 3, \dots, \gamma_j\}$ is connected to $y$ if and only if $y = p_j . \prod_{i \in I \setminus \{j\}} p_i^{y_i}$ for all $y_i \in \{0, 1, 2, \dots, \gamma_i\}$. Clearly, $\deg(p_j) = \prod_{i \in I \setminus \{j\}} (\gamma_i + 1).$

(c) For every prime $p_j|x$, we will have one of these two cases.

**Case 1:** If $p_j|x$ and $j \in B_x$, then the vertex $x$ is clearly adjacent to vertices $y = p_j^{y_j} \cdot \prod_{i \in A_x} p_i^{y_i}$ for all $y_j \in \{1, 2, \ldots, \gamma_j\}$ and for $i \in A_x$, $y_i \in \{0, 1, 2, \ldots, \gamma_i\}$. Hence $x$ is adjacent to $(\gamma_j) \prod_{i \in A_x} (\gamma_i + 1)$ number of vertices for every $j \in B_x$.

**Case 2:** If $p_j|x$ and $j \in C_x$, then $x$ is adjacent to $y = p_j \cdot \prod_{i \in A_x} p_i^{y_i}$ for all $y_i \in \{0, 1, 2, \ldots, \gamma_i\}$. Hence $x$ is adjacent to $\prod_{i \in A_x} (\gamma_i + 1)$ number of vertices for every $j \in C_x$.

Since both cases are exclusively independent, therefore, the required degree is the sum of both cases.

$$\deg(x) = \sum_{j \in B_x} \left( (\gamma_j) \prod_{i \in A_x} (\gamma_i + 1) \right) + \sum_{j \in C_x} \left( \prod_{i \in A_x} (\gamma_i + 1) \right) = \left( \sum_{j \in B_x} (\gamma_j) + \sum_{j \in C_x} (1) \right)$$
$$\cdot \prod_{i \in A_x} (\gamma_i + 1) = \left( \sum_{j \in B_x} \gamma_j + |C_x| \right) \cdot \prod_{i \in A_x} (\gamma_i + 1).$$

□

**Corollary 2.3.** *For any positive integer $\aleph$ if $\Im(p^\aleph q)$ be a common prime divisor graphs, where $p$ and $q$ are distinct prime numbers, then the $\deg(\nu)$ of each vertex $\nu$ of the common prime divisor graph $\Im(p^\aleph q)$ is,*

- $\deg(p) = 2\aleph - 1$ and $\deg(q) = \aleph$.
- $\deg(pq) = \aleph + 1$.
- $\deg(p^\mathfrak{a}) = \deg(p^\mathfrak{a}q) = 2$ if $\mathfrak{a} \in \{2, 3, \ldots, \aleph\}$.

In the following results, we will compute degree-based topological descriptors for common prime divisor graphs $\Im(p^\aleph q)$ using the data from the vertex partition and the edge partitions with degree of vertices. In [37,38], the authors defined the various topological indices based of degrees that contribute intensively in studies of QSAR and QSPR [39–41].

During the proof of following results we may simply denote common prime divisor graph $\Im(p^\aleph q)$ by $\Im$.

**Lemma 2.4.** *Let $\Im(p^\aleph q)$ be a common prime divisor graphs. Then $\mathfrak{T}(\Im) = (\aleph - 1)\left( \widehat{\psi}(2, \aleph) + \widehat{\psi}(2, \aleph + 1) + 2\widehat{\psi}(2, 2\aleph - 1) \right) + \widehat{\psi}(\aleph, \aleph + 1) + \widehat{\psi}(\aleph + 1, 2\aleph - 1).$*

*Proof:* The common prime divisor graph $\Im$ has order $2\aleph + 1$ and the size $4\aleph - 2$. Each vertex of the common prime divisor graph $\Im$ has degree $2$, $\aleph$, $\aleph + 1$, or $2\aleph - 1$, vertices of $\Im$ be partitioned with respect to their degrees. The degree of vertices are given in Corollary 2.3 as:

$$d_\Im(\mathfrak{p}^\mathfrak{a}) = \begin{cases} 2\aleph - 1, & \text{for } \mathfrak{a} = 1 \\ 2, & \text{for } 2 \leq \mathfrak{a} \leq \aleph \end{cases} \tag{11}$$

$$d_\Im(\mathfrak{p}^\mathfrak{a}q) = \begin{cases} \aleph, & \text{for } \mathfrak{a} = 0 \\ \aleph + 1, & \text{for } \mathfrak{a} = 1 \\ 2, & \text{for } 2 \leq \mathfrak{a} \leq \aleph \end{cases} \tag{12}$$

Let

$$\mathcal{V}_\Im(i) = \{\alpha \in \mathcal{V}(\Im) : d_\Im(\alpha) = i\}.$$

It implies $\mathcal{V}_3(i)$ is containing all the those vertices that have degree $i$. Correspondingly, we classify all;

$$\mathcal{V}_3(2) = \{\alpha \in \mathcal{V}(\mathfrak{Z}) : d_3(\alpha) = 2\},$$

$$\mathcal{V}_3(\aleph) = \{\alpha \in \mathcal{V}(\mathfrak{Z}) : d_3(\alpha) = \aleph\},$$

$$\mathcal{V}_3(\aleph + 1) = \{\alpha \in \mathcal{V}(\mathfrak{Z}) : d_3(\alpha) = \aleph + 1\},$$

$$\mathcal{V}_3(2\aleph - 1) = \{\alpha \in \mathcal{V}(\mathfrak{Z}) : d_3(\alpha) = 2\aleph - 1\}.$$

From Eqs (11)–(12), we get $|\mathcal{V}_3(2)| = 2\aleph - 2$, $|\mathcal{V}_3(\aleph)| = 1$, $|\mathcal{V}_3(\aleph + 1)| = 1$ and $|\mathcal{V}_3(2\aleph - 1)| = 1$. In the following Eq (13), the desired edge partition is given.

$$\mathcal{E}_{(\mathfrak{r},\mathfrak{s})}(\mathfrak{Z}) = \begin{cases} \aleph - 1, & \text{for } \mathfrak{r} = 2 \text{ and } \mathfrak{s} = \aleph \\ \aleph - 1, & \text{for } \mathfrak{r} = 2 \text{ and } \mathfrak{s} = \aleph + 1 \\ 2\aleph - 2, & \text{for } \mathfrak{r} = 2 \text{ and } \mathfrak{s} = 2\aleph - 1 \\ 1, & \text{for } \mathfrak{r} = \aleph \text{ and } \mathfrak{s} = \aleph + 1 \\ 1, & \text{for } \mathfrak{r} = \aleph + 1 \text{ and } \mathfrak{s} = 2\aleph - 1. \end{cases} \tag{13}$$

Note that, $\mathcal{E}(\mathfrak{Z}) = \mathcal{E}_{(2,\aleph)}(\mathfrak{Z}) \sqcup \mathcal{E}_{(2,\aleph+1)}(\mathfrak{Z}) \sqcup \mathcal{E}_{(2,2\aleph-1)}(\mathfrak{Z}) \sqcup \mathcal{E}_{(\aleph,\aleph+1)}(\mathfrak{Z}) \sqcup \mathcal{E}_{(\aleph+1,2\aleph+1)}(\mathfrak{Z})$. The cardinality of all edges incident with degree 2 and to a vertex with degree $\aleph$, $\aleph + 1$, $2\aleph - 1$ are precisely $\aleph - 1$, $\aleph - 1$, $2\aleph - 2$, respectively. So $|\mathcal{E}_{(2,\aleph)}(\mathfrak{Z})| = \aleph - 1$, $|\mathcal{E}_{(2,\aleph+1)}(\mathfrak{Z})| = \aleph - 1$, $|\mathcal{E}_{(2,2\aleph-1)}(\mathfrak{Z})| = 2\aleph - 2$. The cardinality of all edges incident with degree $\aleph$ and to a vertex with degree $\aleph + 1$ is exactly 1. Therefore, $|\mathcal{E}_{(\aleph,\aleph+1)}(\mathfrak{Z})| = 1$. Similarly, he cardinality of all edges incident with degree $\aleph + 1$ and to a vertex with degree $2\aleph - 1$ is also 1. Therefore, $|\mathcal{E}_{(\aleph+1,2\aleph-1)}(\mathfrak{Z})| = 1$.

Hence,

$$\begin{aligned} \mathfrak{T}(\mathfrak{Z}) &= \sum_{\alpha\beta \in \mathcal{E}(\mathfrak{Z})} \widehat{\psi}(d_3(\alpha), d_3(\beta)) \\ &= \sum_{\alpha\beta \in \mathcal{E}_{(2,\aleph)}(\mathfrak{Z})} \widehat{\psi}(2, \aleph) + \sum_{\alpha\beta \in \mathcal{E}_{(2,\aleph+1)}(\mathfrak{Z})} \widehat{\psi}(2, \aleph + 1) + \sum_{\alpha\beta \in \mathcal{E}_{(2,2\aleph-1)}(\mathfrak{Z})} \widehat{\psi}(2, 2\aleph - 1) \\ &+ \sum_{\alpha\beta \in \mathcal{E}_{(\aleph,\aleph+1)}(\mathfrak{Z})} \widehat{\psi}(\aleph, \aleph + 1) + \sum_{\alpha\beta \in \mathcal{E}_{(\aleph+1,2\aleph-1)}(\mathfrak{Z})} \widehat{\psi}(\aleph + 1, 2\aleph - 1) \\ &= (\aleph - 1)\widehat{\psi}(2, \aleph) + (\aleph - 1)\widehat{\psi}(2, \aleph + 1) + (2\aleph - 2)\widehat{\psi}(2, 2\aleph - 1) \\ &+ (1)\widehat{\psi}(\aleph, \aleph + 1) + (1)\widehat{\psi}(\aleph + 1, 2\aleph - 1) \\ &= (\aleph - 1)\left(\widehat{\psi}(2, \aleph) + \widehat{\psi}(2, \aleph + 1) + 2\widehat{\psi}(2, 2\aleph - 1)\right) + \widehat{\psi}(\aleph, \aleph + 1) + \widehat{\psi}(\aleph + 1, 2\aleph - 1) \end{aligned}$$

□

Theorems given below determine the degree-based topological descriptors of common prime divisor graphs $\mathfrak{Z}(p^{\aleph}q)$.

**Theorem 2.5.** Let $\mathfrak{Z}(p^{\aleph}q)$ be a common prime divisor graphs. Then the $\mathfrak{R}_{\alpha}$ index or the general Randić index of the common prime divisor graph $\mathfrak{Z}(p^{\aleph}q)$ is;

$$\mathfrak{R}_{\alpha}(\mathfrak{Z}) = (\aleph - 1)\left((2\aleph)^{\alpha} + (2\aleph + 2)^{\alpha} + 2(4\aleph - 2)^{\alpha}\right) + \aleph^{\alpha}(\aleph + 1)^{\alpha} + (\aleph + 1)^{\alpha}(2\aleph - 1)^{\alpha},$$

the Randić index of the common prime divisor graph $\mathfrak{Z}(p^{\aleph}q)$ is;

$$\mathfrak{R}_{-\frac{1}{2}}(\mathfrak{Z}) = \frac{\aleph-1}{\sqrt{2}} \left( \frac{1}{\sqrt{\aleph}} + \frac{1}{\sqrt{\aleph+1}} + \frac{2}{\sqrt{2\aleph-1}} \right) + \frac{1}{\sqrt{\aleph(\aleph+1)}} + \frac{1}{\sqrt{(\aleph+1)(2\aleph-1)}},$$

the second Zagreb index of the common prime divisor graph $\mathfrak{Z}(p^{\aleph}q)$ is;

$$\mathfrak{R}_1(\mathfrak{Z}) = 15\aleph^2 - 12\aleph + 1,$$

the second modified Zagreb index of the common prime divisor graph $\mathfrak{Z}(p^{\aleph}q)$ is;

$$\mathfrak{R}_{-1}(\mathfrak{Z}) = \frac{6\aleph^3 - 4\aleph^2 + 3\aleph - 1}{4\aleph^3 + 2\aleph^2 - 2\aleph},$$

the $\chi_{\alpha}$ index or the general sum-connectivity index of the common prime divisor graph $\mathfrak{Z}(p^{\aleph}q)$ is;

$$\chi_{\alpha}(\mathfrak{Z}) = (\aleph-1)\left((\aleph+3)^{\alpha} + (\aleph+2)^{\alpha} + 2(2\aleph+1)^{\alpha}\right) + (2\aleph+1)^{\alpha} + (3\aleph)^{\alpha},$$

the sum-connectivity index of the common prime divisor graph $\mathfrak{Z}(p^{\aleph}q)$ is;

$$\chi_{-\frac{1}{2}}(\mathfrak{Z}) = \frac{18\aleph^4 + 53\aleph^3 + 11\aleph^2 - 16\aleph + 6}{3\aleph(2\aleph+1)(\aleph+3)(\aleph+2)},$$

the first Zagreb index of the common prime divisor graph $\mathfrak{Z}(p^{\aleph}q)$ is;

$$\chi_1(\mathfrak{Z}) = 6\aleph^2 + 6\aleph - 6,$$

and the hyper-Zagreb index of the common prime divisor graph $\mathfrak{Z}(p^{\aleph}q)$ is;

$$\chi_2(\mathfrak{Z}) = 10\aleph^3 + 21\aleph^2 + \aleph - 14.$$

*Proof*: In order to find the general Randić index $R_{\alpha}$ of $\mathfrak{Z}$, we obtain $\widehat{\psi}(d_{\mathfrak{Z}}(\gamma), d_{\mathfrak{Z}}(\beta))$
$= \left(d_{\mathfrak{Z}}(\gamma) \times d_{\mathfrak{Z}}(\beta)\right)^{\alpha}$.

So $\widehat{\psi}(2,\aleph) = (2\aleph)^{\alpha}, \widehat{\psi}(2,\aleph+1) = (2\aleph+2)^{\alpha}, \widehat{\psi}(2,2\aleph-1) = (4\aleph-2)^{\alpha}, \widehat{\psi}(\aleph,\aleph+1)$
$= \aleph^{\alpha}(\aleph+1)^{\alpha}, \widehat{\psi}(\aleph+1,2\aleph-1) = (\aleph+1)^{\alpha}(2\aleph-1)^{\alpha}$.

Thus by Lemma 2.4,

$\mathfrak{R}_{\alpha}(\mathfrak{Z}) = (\aleph-1)\left(\widehat{\psi}(2,\aleph) + \widehat{\psi}(2,\aleph+1) + 2\widehat{\psi}(2,2\aleph-1)\right) + \widehat{\psi}(\aleph,\aleph+1) + \widehat{\psi}(\aleph+1,2\aleph-1)$
$= (\aleph-1)\left((2\aleph)^{\alpha} + (2\aleph+2)^{\alpha} + 2(4\aleph-2)^{\alpha}\right) + \aleph^{\alpha}(\aleph+1)^{\alpha} + (\aleph+1)^{\alpha}(2\aleph-1)^{\alpha}$,

For $\alpha = -\frac{1}{2}$, we obtain the Randić index;

$$\mathfrak{R}_{-\frac{1}{2}}(\mathfrak{Z}) = \frac{\aleph-1}{\sqrt{2}} \left( \frac{1}{\sqrt{\aleph}} + \frac{1}{\sqrt{\aleph+1}} + \frac{2}{\sqrt{2\aleph-1}} \right) + \frac{1}{\sqrt{\aleph(\aleph+1)}} + \frac{1}{\sqrt{(\aleph+1)(2\aleph-1)}}.$$

Also, for $\alpha = 1$, the second Zagreb index is; $\mathfrak{R}_1(\mathfrak{Z}) = (\aleph-1)(2\aleph + (2\aleph+2) + 2(4\aleph-2)) + \aleph(\aleph+1) + (\aleph+1)(2\aleph-1) = 15\aleph^2 - 12\aleph + 1$.

Similarly, the second modified Zagreb index for $\alpha = -1$ is;

$$\mathfrak{R}_{-1}(\mathfrak{Z}) = \frac{\aleph - 1}{2}\left(\frac{1}{\aleph} + \frac{1}{\aleph + 1} + \frac{2}{2\aleph - 1}\right) + \frac{1}{\aleph(\aleph + 1)} + \frac{1}{(\aleph + 1)(2\aleph - 1)}$$

$$= \frac{6\aleph^3 - 4\aleph^2 + 3\aleph - 1}{4\aleph^3 + 2\aleph^2 - 2\aleph}.$$

We can obtain the index related to general sum-connectivity as; $\chi_\alpha$ of $\mathfrak{Z}$, we get $\widehat{\psi}(d_{\mathfrak{Z}}(\gamma), d_{\mathfrak{Z}}(\beta)) = (d_{\mathfrak{Z}}(\gamma) + d_{\mathfrak{Z}}(\beta))^\alpha$. So $\widehat{\psi}(2, \aleph) = (\aleph + 2)^\alpha, \widehat{\psi}(2, \aleph + 1) = (\aleph + 3)^\alpha$, $\widehat{\psi}(2, 2\aleph - 1) = (2\aleph + 1)^\alpha, \widehat{\psi}(\aleph, \aleph + 1) = (2\aleph + 1)^\alpha, \widehat{\psi}(\aleph + 1, 2\aleph - 1) = (3\aleph)^\alpha$.

Thus by Lemma 2.4, $\chi_\alpha(\mathfrak{Z}) = (\aleph - 1)\left(\widehat{\psi}(2, \aleph) + \widehat{\psi}(2, \aleph + 1) + 2\widehat{\psi}(2, 2\aleph - 1)\right) + \widehat{\psi}(\aleph, \aleph + 1) + \widehat{\psi}(\aleph + 1, 2\aleph - 1) = (\aleph - 1)\left((\aleph + 2)^\alpha + (\aleph + 3)^\alpha + 2(2\aleph + 1)^\alpha\right) + (2\aleph + 1)^\alpha + (3\aleph)^\alpha$.

If $\alpha = -\frac{1}{2}$, then the index related to sum-connectivity is,

$$\chi_{-\frac{1}{2}}(\mathfrak{Z}) = (\aleph - 1)\left(\frac{1}{\aleph + 2} + \frac{1}{\aleph + 3} + \frac{2}{2\aleph + 1}\right) + \frac{1}{2\aleph + 1} + \frac{1}{3\aleph}$$

$$= \frac{18\aleph^4 + 53\aleph^3 + 11\aleph^2 - 16\aleph + 6}{3\aleph\,(2\aleph + 1)\,(\aleph + 3)\,(\aleph + 2)}.$$

If $\alpha = 1$, then we get the first Zagreb index $\chi_1(\mathfrak{Z})$ as;

$$\chi_1(\mathfrak{Z}) = (\aleph - 1)\left((\aleph + 3) + (\aleph + 2) + 2(2\aleph + 1)\right) + (2\aleph + 1) + (3\aleph) = 6\aleph^2 + 6\aleph - 6.$$

In the case if $\alpha = 2$, then the index related to hyper-Zagreb $\chi_2(\mathfrak{Z})$ is;

$$\chi_2(\mathfrak{Z}) = (\aleph - 1)\left((\aleph + 2)^2 + (\aleph + 3)^2 + 2(2\aleph + 1)^2\right) + (2\aleph + 1)^2 + (3\aleph)^2$$

$$= 10\aleph^3 + 21\aleph^2 + \aleph - 14.$$

$\square$

**Theorem 2.6.** *Let $\mathfrak{Z}(p^\aleph q)$ be a common prime divisor graphs. Then the GA index of the graph $\mathfrak{Z}(p^\aleph q)$ is;*

$$GA(\mathfrak{Z}) = (\aleph - 1)\left(\frac{2\sqrt{2\aleph}}{\aleph + 2} + \frac{2\sqrt{2\aleph + 2}}{\aleph + 3} + \frac{4\sqrt{4\aleph - 2}}{2\aleph + 1}\right) + \frac{2\sqrt{\aleph(\aleph + 1)}}{2\aleph + 1} + \frac{2\sqrt{(\aleph + 1)(2\aleph - 1)}}{3\aleph},$$

*the ABC index of the graph $\mathfrak{Z}(p^\aleph q)$ is;*

$$ABC(\mathfrak{Z}) = 2\sqrt{2}\,(\aleph - 1) + \sqrt{\frac{2\aleph - 1}{\aleph(\aleph + 1)}} + \sqrt{\frac{3\aleph - 2}{(\aleph + 1)(2\aleph - 1)}},$$

*the AZI index or the augmented Zagreb index of the graph $\mathfrak{Z}(p^\aleph q)$ is;*

$$AZI(\mathfrak{Z}) = 32\,(\aleph - 1) + \frac{\aleph^3(\aleph + 1)^3}{(2\aleph - 1)^3} + \frac{(\aleph + 1)^3(2\aleph - 1)^3}{(3\aleph - 2)^3},$$

*the H index or the harmonic index of the graph $\mathfrak{Z}(p^\aleph q)$ is;*

$$H(\mathfrak{Z}) = \frac{36\aleph^4 + 106\aleph^3 + 22\aleph^2 - 32\aleph + 12}{3\aleph\,(\aleph+2)\,(\aleph+3)\,(2\aleph+1)},$$

*the SDD index of the graph $\mathfrak{Z}(p^\aleph q)$ is;*

$$SDD(\mathfrak{Z}) = \frac{12\aleph^5 - 8\aleph^4 + 31\aleph^3 - 8\aleph^2 - 9\aleph + 2}{4\aleph^3 + 2\aleph^2 - 2\aleph}.$$

*Proof*: The *GA* or the geometric-arithmetic index of the common prime divisor graph $\mathfrak{Z}$, we get, $\widehat{\psi}(d_3(\gamma), d_3(\beta)) = \frac{2\sqrt{d_3(\gamma) \times d_3(\beta)}}{d_3(\gamma) + d_3(\beta)}$. So $\widehat{\psi}(2, \aleph) = \frac{2\sqrt{2\aleph}}{\aleph+2}$, $\widehat{\psi}(2, \aleph+1) = \frac{2\sqrt{2\aleph+2}}{\aleph+3}$, $\widehat{\psi}(2, 2\aleph-1) = \frac{2\sqrt{4\aleph-2}}{2\aleph+1}$, $\widehat{\psi}(\aleph, \aleph+1) = \frac{2\sqrt{\aleph(\aleph+1)}}{2\aleph+1}$ and $\widehat{\psi}(\aleph+1, 2\aleph-1) = \frac{2\sqrt{(\aleph+1)(2\aleph-1)}}{3\aleph}$.

Thus by Lemma 2.4, $GA(\mathfrak{Z}) = (\aleph - 1)\left(\frac{2\sqrt{2\aleph}}{\aleph+2} + \frac{2\sqrt{2\aleph+2}}{\aleph+3} + \frac{4\sqrt{4\aleph-2}}{2\aleph+1}\right) + \frac{2\sqrt{\aleph(\aleph+1)}}{2\aleph+1} + \frac{2\sqrt{(\aleph+1)(2\aleph-1)}}{3\aleph}$.

For the *ABC* index of the common prime divisor graph $\mathfrak{Z}$, we obtained

$$\widehat{\psi}(d_3(\gamma), d_3(\beta)) = \sqrt{\frac{d_3(\gamma) + d_3(\beta) - 2}{d_3(\gamma) \times d_3(\beta)}}.$$

So $\widehat{\psi}(2, \aleph) = \sqrt{\frac{1}{2}}$, $\widehat{\psi}(2, \aleph+1) = \sqrt{\frac{1}{2}}$, $\widehat{\psi}(2, 2\aleph-1) = \sqrt{\frac{1}{2}}$, $\widehat{\psi}(\aleph, \aleph+1) = \sqrt{\frac{2\aleph-1}{\aleph(\aleph+1)}}$ and $\widehat{\psi}(\aleph+1, 2\aleph-1) = \sqrt{\frac{3\aleph-2}{(\aleph+1)(2\aleph-1)}}$. Thus by Lemma 2.4, $ABC(\mathfrak{Z}) = (\aleph-1)\left(\sqrt{\frac{1}{2}} + \sqrt{\frac{1}{2}} + 2\sqrt{\frac{1}{2}}\right) + \sqrt{\frac{2\aleph-1}{\aleph(\aleph+1)}} + \sqrt{\frac{3\aleph-2}{(\aleph+1)(2\aleph-1)}} = 2\sqrt{2}\,(b-1) + \sqrt{\frac{2\aleph-1}{\aleph(\aleph+1)}} + \sqrt{\frac{3\aleph-2}{(\aleph+1)(2\aleph-1)}}$.

For the *AZI* index or the augmented Zagreb index of the common prime divisor graph $\mathfrak{Z}$, we obtain $\widehat{\psi}(d_3(\gamma), d_3(\beta)) = \left(\frac{d_3(\gamma) \times d_3(\beta)}{d_3(\gamma) + d_3(\beta) - 2}\right)^3$. So $\widehat{\psi}(2, \aleph) = 8$, $\widehat{\psi}(2, \aleph+1) = 8$, $\widehat{\psi}(2, 2\aleph-1) = 8$, $\widehat{\psi}(\aleph, \aleph+1) = \left(\frac{\aleph(\aleph+1)}{2\aleph-1}\right)^3$ and $\widehat{\psi}(\aleph+1, 2\aleph-1) = \left(\frac{(\aleph+1)(2\aleph-1)}{3\aleph-2}\right)^3$. Thus by Lemma 2.4,

$$AZI(\mathfrak{Z}) = (\aleph-1)(8 + 8 + 16) + \left(\frac{\aleph(\aleph+1)}{2\aleph-1}\right)^3 + \left(\frac{(\aleph+1)(2\aleph-1)}{3\aleph-2}\right)^3$$

$$= 32\,(\aleph-1) + \frac{\aleph^3(\aleph+1)^3}{(2\aleph-1)^3} + \frac{(\aleph+1)^3(2\aleph-1)^3}{(3\aleph-2)^3}.$$

For the *H* index or the harmonic index of the common prime divisor graph $\mathfrak{Z}$, we obtain $\widehat{\psi}(d_3(\gamma), d_3(\beta)) = \frac{2}{d_3(\gamma) + d_3(\beta)}$.

So $\widehat{\psi}(2, \aleph) = \frac{2}{\aleph+2}$, $\widehat{\psi}(2, \aleph+1) = \frac{2}{\aleph+3}$, $\widehat{\psi}(2, 2\aleph-1) = \frac{2}{2\aleph+1}$, $\widehat{\psi}(\aleph, \aleph+1) = \frac{2}{2\aleph+1}$ and $\widehat{\psi}(\aleph+1, 2\aleph-1) = \frac{2}{3\aleph}$. Thus by Lemma 2.4,

$$H(\mathfrak{Z}) = (\aleph-1)\left(\frac{2}{\aleph+2} + \frac{2}{\aleph+3} + \frac{4}{2\aleph+1}\right) + \frac{2}{2\aleph+1} + \frac{2}{3\aleph}$$

$$= \frac{36\aleph^4 + 106\aleph^3 + 22\aleph^2 - 32\aleph + 12}{3\aleph\,(\aleph+2)\,(\aleph+3)\,(2\aleph+1)}.$$

For the *SDD* index or the symmetric division degree index of the common prime divisor graph $\mathfrak{Z}(p^\aleph q)$ we obtain $\widehat{\psi}(d_3(\gamma), d_3(\beta)) = \frac{(d_3(\gamma))^2 + (d_3(\beta))^2}{d_3(\gamma) \times d_3(\beta)}$.

So $\widehat{\psi}(2,\aleph) = \frac{\aleph^2+4}{2\aleph}$, $\widehat{\psi}(2,\aleph+1) = \frac{(\aleph+1)^2+4}{2\aleph+2}$, $\widehat{\psi}(2,2\aleph-1) = \frac{(2\aleph-1)^2+4}{4\aleph-2}$, $\widehat{\psi}(\aleph,\aleph+1) = \frac{\aleph^2+(\aleph+1)^2}{\aleph(\aleph+1)}$ and $\widehat{\psi}(\aleph+1,2\aleph-1) = \frac{(\aleph+1)^2+(2\aleph-1)^2}{(\aleph+1)(2\aleph-1)}$.

Thus by Lemma 2.4,

$$SDD(\mathfrak{Z}) = (\aleph-1)\left(\frac{\aleph^2+4}{2\aleph} + \frac{(\aleph+1)^2+4}{2\aleph+2} + \frac{(2\aleph-1)^2+4}{2\aleph-1}\right)$$
$$+ \frac{\aleph^2+(\aleph+1)^2}{\aleph(\aleph+1)} + \frac{(\aleph+1)^2+(2\aleph-1)^2}{(\aleph+1)(2\aleph-1)}$$

$$= \frac{12\aleph^5 - 8\aleph^4 + 31\aleph^3 - 8\aleph^2 - 9\aleph + 2}{4\aleph^3 + 2\aleph^2 - 2\aleph}.$$

$\square$

**Theorem 2.7.** *Let $\mathfrak{Z}(p^\aleph q)$ be a common prime divisor graphs. Then the $ReZG_1$ index of the graph $\mathfrak{Z}(p^\aleph q)$ is; $ReZG_1(\mathfrak{Z}) = 1 + 2\aleph$, the $ReZG_2$ index of the graph $\mathfrak{Z}(p^\aleph q)$ is;*

$$ReZG_2(\mathfrak{Z}) = \frac{55\aleph^5 + 186\aleph^4 + 28\aleph^3 - 168\aleph^2 + 49\aleph - 6}{3\aleph\,(1+2\aleph)\,(3+\aleph)\,(2+\aleph)},$$

*the $ReZG_3$ index of the graph $\mathfrak{Z}(p^\aleph q)$ is;*

$$ReZG_3(\mathfrak{Z}) = 42\aleph^3 + 7\aleph^2 - 18\aleph + 1.$$

*Proof*: For the $ReZG_1$ index or first redefined Zagreb index of the common prime divisor graph $\mathfrak{Z}(p^\aleph q)$ we obtain $\widehat{\psi}(d_3(\gamma), d_3(\beta)) = \frac{d_3(\gamma)+d_3(\beta)}{d_3(\gamma)\times d_3(\beta)}$. So $\widehat{\psi}(2,\aleph) = \frac{2+\aleph}{2\aleph}$, $\widehat{\psi}(2,\aleph+1) = \frac{3+\aleph}{2\aleph+2}$, $\widehat{\psi}(2,2\aleph-1) = \frac{1+2\aleph}{4\aleph-2}$, $\widehat{\psi}(\aleph,\aleph+1) = \frac{2\aleph+1}{\aleph^2+\aleph}$ and $\widehat{\psi}(\aleph+1,2\aleph-1) = \frac{3\aleph}{2\aleph^2+\aleph-1}$. Thus by Lemma 2.4, $ReZG_1(\mathfrak{Z}) = 1 + 2\aleph$.

For the $ReZG_2$ index or the second redefined Zagreb index of the common prime divisor graph $\mathfrak{Z}$, we obtain $\widehat{\psi}(d_3(\gamma), d_3(\beta)) = \frac{d_3(\gamma)\times d_3(\beta)}{d_3(\gamma)+d_3(\beta)}$. So $\widehat{\psi}(2,\aleph) = \frac{2\aleph}{2+\aleph}$, $\widehat{\psi}(2,\aleph+1) = \frac{2\aleph+2}{3+\aleph}$, $\widehat{\psi}(2,2\aleph-1) = \frac{4\aleph-2}{1+2\aleph}$, $\widehat{\psi}(\aleph,\aleph+1) = \frac{\aleph^2+\aleph}{2\aleph+1}$, finally $\widehat{\psi}(\aleph+1,2\aleph-1) = \frac{2\aleph^2+\aleph-1}{3\aleph}$.

Thus by Lemma 2.4, $ReZG_2(\mathfrak{Z}) = \frac{55\aleph^5+186\aleph^4+28\aleph^3-168\aleph^2+49\aleph-6}{3\aleph(1+2\aleph)(3+\aleph)(2+\aleph)}$.

For the $ReZG_3$ index or the third redefined Zagreb index of the common prime divisor graph $\mathfrak{Z}$, we obtain $\widehat{\psi}(d_3(\gamma), d_3(\beta)) = \big(d_3(\gamma) \times d_3(\beta)\big)\big(d_3(\gamma) + d_3(\beta)\big)$. So $\widehat{\psi}(2,\aleph) = (2\aleph)(2+\aleph)$, $\widehat{\psi}(2,\aleph+1) = (2\aleph+2)(3+\aleph)$, $\widehat{\psi}(2,2\aleph-1) = (4\aleph-2)(1+2\aleph)$, $\widehat{\psi}(\aleph,\aleph+1) = (\aleph^2+\aleph)(2\aleph+1)$ and $\widehat{\psi}(\aleph+1,2\aleph-1) = (2\aleph^2+\aleph-1)(3\aleph)$. Thus by Lemma 2.4, $ReZG_3(\mathfrak{Z}) = \aleph(1+2\aleph)^2 + (8\aleph-7)\aleph(1+2\aleph) + (2\aleph-2)\aleph(2+\aleph) + (2\aleph-2)\aleph(3+\aleph) + 2\aleph(3\aleph)^2 - 2\aleph + \aleph(3\aleph) + 1 = 42\aleph^3 + 7\aleph^2 - 18\aleph + 1$. $\square$

## 3 Conclusion

In this article, we have defined a new class of graphs associated with positive integers called the common prime divisor graphs. We have also explored algebraic formulations and novel topological descriptors related to common prime divisor graphs. The methodology begins by defining the graph for common prime divisor of a fix positive integer and a few results to elucidate the algebraic properties, specifically addressing topological indices function. Subsequently, leveraging those results, we have studied a few degree-based indices for various parameters, including alpha, geometric arithmetic, atom-bond connectivity, augmented

Zagreb, Albertson, and redefined Zagreb indices, applied specifically to common prime divisor graphs. This study remains opens to explore for other topological indices and graph invariant associated with the different classes of common prime divisor graphs associated to positive integers.

## Acknowledgments

The authors gratefully acknowledge the funding of the Deanship of Graduate Studies and Scientific Research, Jazan University, Saudi Arabia, through project number: RG24-L04.

## Author contributions

**Conceptualization:** Ali N. A. Koam.

**Data curation:** Ali N. A. Koam.

**Formal analysis:** Azeem Haider.

**Funding acquisition:** Ali N. A. Koam.

**Investigation:** Ali Ahmad.

**Methodology:** Moin Akhtar Ansari.

**Project administration:** Ali N. A. Koam.

**Resources:** Moin Akhtar Ansari.

**Software:** Azeem Haider.

**Supervision:** Azeem Haider.

**Validation:** Moin Akhtar Ansari.

**Visualization:** Ali Ahmad.

**Writing – original draft:** Ali N. A. Koam, Azeem Haider, Ali Ahmad, Moin Akhtar Ansari.

**Writing – review & editing:** Ali N. A. Koam, Azeem Haider, Ali Ahmad, Moin Akhtar Ansari.

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
