## [Decision Letter · Decision Letter 0]

6 Feb 2025

PONE-D-24-59163A novel approach to explore Common Prime Divisor Graphs and their degree based topological descriptorPLOS ONE

Dear Dr. Ansari,

Thank you for submitting your manuscript to PLOS ONE. After careful consideration, we feel that it has merit but does not fully meet PLOS ONE’s publication criteria as it currently stands. Therefore, we invite you to submit a revised version of the manuscript that addresses the points raised during the review process.

We look forward to receiving your revised manuscript.

Kind regards,

Deepak Singh, Ph D

Academic Editor

PLOS ONE

Journal Requirements:

 “Deputyship for Research \& Innovation, Ministry of Education in Saudi Arabia for funding this research work through project number ISP- 2024.”

“All authors declare no conflicts of interest in this paper.”

Reviewers' comments:

Reviewer's Responses to Questions

**Comments to the Author**

1. Is the manuscript technically sound, and do the data support the conclusions?

Reviewer #1: Yes

2. Has the statistical analysis been performed appropriately and rigorously? 

Reviewer #1: Yes

3. Have the authors made all data underlying the findings in their manuscript fully available?

Reviewer #1: Yes

4. Is the manuscript presented in an intelligible fashion and written in standard English?

Reviewer #1: Yes

5. Review Comments to the Author

Reviewer #1: The authors have introduced a class of graphs constructed on the divisors of

a positive integers, called the common prime divisor graphs. They have explored

a few fundamental invariants of these graphs for a speci c case. They have also

computed various graph-associated topological indices for such graph. In partic-

ular, they have presented the degree-based indices for parameters such as alpha,

geometric arithmetic, atom-bond connectivity, augmented Zagreb, Albertson, and

rede ned Zagreb indices.

The way of construction of common prime divisor graph is comparatively a novel

approach and has a potential for the future because such graphs can be studied

over various other classes of positive integers. The work presented in this article

is mathematically correct, the paper is well-written and effectively communicates

its outcomes. Some comments about the formatting of the article are given below

that needs to be consider before publication.

1. All equations and Theorem are numbered with 0.1, 0.2, etc. Please correct

the format.

2. There are a few equations that are too long to be adjusted in one line, please

break and adjust accordingly. See proofs of Theorem 0.5 and Theorem 0.6.

3. A general comment: several unnecessary citation are included that can be re-

duced. See for example [13], [15], [18], [38] and [42].

6. PLOS authors have the option to publish the peer review history of their article (what does this mean?). If published, this will include your full peer review and any attached files.

Reviewer #1: No

---

## [Author Response · Author response to Decision Letter 1]

7 Apr 2025

Response to Editor

Dear Editor

Thank you for your valuable comments and suggestions.

Keeping all 6 points in our consideration we have revised the manuscript as per your comments and suggestions.

A file is attached in the metadata with this revised version.

Response to Reviewers

Dear Reviewer

Thank you for your valuable comments and suggestions.

Point Wise Replies to the Reviewers Comments

Comments 1: All equations and Theorem are numbered with 0.1, 0.2, etc. Please correct

the format.

Reply: We have corrected the format as required.

Comments 2: There are a few equations that are too long to be adjusted in one line, please

break and adjust accordingly. See proofs of Theorem 0.5 and Theorem 0.6.

Reply: We have broken the line and adjusted accordingly.

Comments 3: A general comment: several unnecessary citation are included that can be reduced. See for example [13], [15], [18], [38] and [42].

Reply: We have removed unnecessary citations in the revised version and updated it.

Dr. Moin Akhtar Ansari

(Associate Professor)

Dept. of Mathematics

College of Science

Jazan University, P.O. Box. 114

Jazan 45142

Kingdom of Saudi Arabia

---

## [Editor Report · Decision Letter 1]

16 Apr 2025

A novel approach to explore Common Prime Divisor Graphs and their degree based topological descriptor

PONE-D-24-59163R1

Dear Dr. Ansari,

We’re pleased to inform you that your manuscript has been judged scientifically suitable for publication and will be formally accepted for publication once it meets all outstanding technical requirements.

Kind regards,

Deepak Singh, Ph D

Academic Editor

PLOS ONE
---

## [Editor Report · Acceptance letter]

PONE-D-24-59163R1

PLOS ONE

Dear Dr. Ansari,

I'm pleased to inform you that your manuscript has been deemed suitable for publication in PLOS ONE. Congratulations! Your manuscript is now being handed over to our production team.

Kind regards,

on behalf of

Dr. Deepak Singh

Academic Editor

PLOS ONE